# Gut Microbiome in Pulmonary Arterial Hypertension—An Emerging Frontier

**DOI:** 10.3390/idr17030066

**Published:** 2025-06-09

**Authors:** Sasha Z. Prisco, Suellen D. Oliveira, E. Kenneth Weir, Thenappan Thenappan, Imad Al Ghouleh

**Affiliations:** 1Cardiovascular Division, Department of Medicine, University of Minnesota, Minneapolis, MN 55455, USA; weirx002@umn.edu (E.K.W.); tthenapp@umn.edu (T.T.); 2Vascular Immunobiology Lab, Department of Anesthesiology, Department of Physiology and Biophysics, College of Medicine, University of Illinois Chicago, Chicago, IL 60612, USA; suelleno@uic.edu; 3Cardiovascular Research Center, Brown University Health Cardiovascular Institute, Department of Medicine, The Warren Alpert Medical School of Brown University, Providence, RI 02908, USA

**Keywords:** gut microbiome, pulmonary arterial hypertension, inflammation

## Abstract

Pulmonary arterial hypertension (PAH) is an irreversible disease characterized by vascular and systemic inflammation, ultimately leading to right ventricular failure. There is a great need for adjunctive therapies to extend survival for PAH patients. The gut microbiome influences the host immune system and is a potential novel target for PAH treatment. We review the emerging preclinical and clinical evidence which strongly suggests that there is gut dysbiosis in PAH and that alterations in the gut microbiome may either initiate or facilitate the progression of PAH by modifying systemic immune responses. We also outline approaches to modify the intestinal microbiome and delineate some practical challenges that may impact efforts to translate preclinical microbiome findings to PAH patients. Finally, we briefly describe studies that demonstrate contributions of infections to PAH pathogenesis. We hope that this review will propel further investigations into the mechanisms by which gut dysbiosis impacts PAH and/or right ventricular function, approaches to modify the gut microbiome, and the impact of infections on PAH development or progression.

## 1. Introduction

Vascular injury and inflammation are key drivers of pulmonary vascular remodeling in pulmonary arterial hypertension (PAH) [1,2,3,4,5]. Primarily known as a cardiopulmonary disease, growing evidence has implicated interorgan communication in PAH pathogenesis. We and others have observed the potential role of the intestinal microbiome in the onset and progression of PAH. The gastrointestinal system contributes to systemic inflammation as it contains about 70–80% of the body’s immune cells [6,7], along with trillions of microorganisms known as the gut microbiome [8]. The composition of the gut microbiome can be influenced by many factors, including environmental pollutants, medications, nutrient availability/diet, exercise, oxygen level, sex, age, and genetics (Figure 1) [9]. There is now evidence of the gut microbiome’s impact on many chronic diseases [10]. Clinical trials of targeted anti-inflammatory drugs to date have not clearly demonstrated benefits in PAH [11,12,13], partially due to enrollment size, trial design, and patient selection. Modulating a key immune organ, such as the intestine, to alter many inflammatory pathways may be an effective adjunctive approach to treating PAH.

This review describes the experimental (Table 1) and clinical evidence (Table 2) which implicates the gut microbiota as key contributors to PAH pathogenesis. Additionally, we provide a brief overview of the effect of infections on PAH development and the gut microbiome. While there are five broad World Health Organization (WHO) groups of pulmonary hypertension (PH), which are defined by a mean pulmonary artery pressure > 20 mmHg [14], we primarily focus on WHO Group 1 PAH and Group 3 PH (PH secondary to lung disease) in this review. The current overarching hypothesis is that gut dysbiosis leads to increased systemic inflammation and the progression of PAH and/or right ventricular (RV) dysfunction (Figure 2).

## 2. Preclinical Studies of the Gut–Lung Axis in Pulmonary Hypertension

Multiple studies demonstrate that PH rodent models have alterations in the gut microbiome commonly referred to as gut dysbiosis [15,16,17,18,19,20,21,22,23] (Table 1). Monocrotaline (MCT) PAH rats have increased intestinal permeability as measured by fluorescein isothiocyanate (FITC)-dextran, lipopolysaccharide (LPS), and soluble CD14 in the blood [24] and disrupted intestinal morphology, with greater muscularis layer thickness and fibrosis and diminished villus length and goblet cell number [21,25]. Sugen hypoxia (SuHx) rats have an elevated fecal *Firmicutes*-to-*Bacteroidetes* ratio and fewer short-chain fatty acid (SCFA)-producing bacteria, such as those that are acetate- and butyrate-producing, with no change in lactate-producing bacteria [15]. The milder hypoxic rat PH model also has altered and distinct gut microbiota with more arginine and arginine-producing bacteria, *Blautia* and *Bifidobacterium*, and the trimethylamine N-oxide (TMAO) biosynthetic bacteria, *Streptococcus* [19]. Hypoxic mice have disrupted gut microbiome composition with increases in the genera *Prevotella*, *Oscillospira*, and *Ruminococcus* and decreases in *Lactobacillus* [26]. Alterations in the gut microbiome also occur in the large animal, bovine brisket disease PH model, with lower total volatile fatty acids and alpha diversity (richness and evenness of bacteria) in rumen fluid [27]. The specific intestinal microbiome differences observed in various PH rodent models are comprehensively reviewed elsewhere [28].

Several preclinical studies have modified the gut microbiome to assess its effects on PAH. Altering the microbiome with a broad-spectrum antibiotic cocktail (ampicillin, vancomycin, neomycin, and metronidazole) prior to SU5416 administration mitigates PAH in SuHx rats [29]. Additionally, diet impacts PAH pathogenesis and severity. For example, apolipoprotein E (ApoE) knockout mice given a Paigen (high fat, high cholesterol) diet develop PAH [30] compared to ApoE knockout mice fed normal chow. Double ApoE/IL1R1 knockout mice consuming a Paigen diet have even worse PAH, implicating the role of IL-1 in PAH pathogenesis [30]. Western diet increases right ventricular systolic pressure (RVSP) and RV myocardial lipid deposition and reduces RV function in mice [31]. Metformin decreases RVSP and RV lipid and ceramide accumulation [31]. Moreover, a high-soluble-fiber diet attenuates hypoxia-induced pulmonary vascular remodeling by increasing the abundance of SCFA-producing bacteria (*Bacteroides*, *Anaerostripes*, and *Anaerocolumna*), diminishing proinflammatory bacteria (*Romboutsia*, *Mammalicoccus*, *Staphylococcus*, *Clostridioides*, and *Streptococcus*), and reducing lung interstitial macrophages, dendritic cells, and nonclassical monocytes [32]. The serum metabolites, phosphatidylcholines, lysophosphatidylcholines, ceramides, and hexosylceramides, are lower, while propionylcarnitine and probetaine are greater, in high-soluble-fiber-fed mice compared to low-soluble-fiber-fed hypoxic mice [32]. Interestingly, treatment with a phosphodiesterase-5 inhibitor (tadalafil) or an endothelin pathway inhibitor (macitentan) in SuHx rats reduces the plasma levels of many phosphatidylcholines [33]. However, not all suspected advantageous dietary interventions have been effective in PAH. Although intermittent fasting augments RV function and extends survival in MCT rats, there is minimal effect on PAH severity [25], suggesting that more targeted dietary interventions may be needed to alter PAH severity.

One of the primary ways that the gut microbiota interact with their host is via metabolites, which are intermediate or end products of microbial metabolism [34]. The gut metabolites are generated from bacterial metabolism of dietary or other host substrates. Supplementation with the SCFA, butyrate, which is an endogenous histone deacetylase (HDAC) inhibitor, attenuates pulmonary vascular remodeling and accumulation of alveolar (CD68+) and interstitial (CD68+ and CD163+) lung macrophages in hypoxic PH [35]. Butyrate’s positive effects on pulmonary vascular remodeling have been corroborated in other unpublished studies [36,37] that suggest that butyrate also regulates endothelial cell inflammatory activation and migration. However, the improvement in PH with butyrate treatment appears to only be seen in prevention models but not when given after two and four weeks of hypoxic exposure [35].

Trimethylamine N-oxide (TMAO), derived from trimethylamine (TMA), is a bacterial metabolite generated by the breakdown of dietary choline, carnitine, and betaine. It is elevated in intermediate- and high-risk idiopathic PAH (IPAH) patients [38]. Administering TMAO to hypoxic mice worsens PH through macrophage secretion [38]. Treating hypoxic and MCT rodents with a structural analog of choline that inhibits TMAO synthesis, 3,3-dimethyl-1-butanol (DMB), reduces RV systolic pressure and pulmonary vascular thickness/muscularization and suppresses cytokine and chemokine signaling, with the strongest associations with *Cxcl6* and *Il6* [38,39]. While TMAO worsens PAH, long-term TMAO treatment of MCT rats may be beneficial to RV function by preserving fatty acid oxidation and decreasing pyruvate metabolism, thus preserving mitochondrial energy metabolism and mitigating the development of RV dysfunction in PAH [40]. The role of other microbial metabolites [34,40,41,42], such as amino acid metabolites and retinoic and bile acids in PAH and RV failure is not well-defined.

There are few publications investigating the outcomes of direct manipulation of the microbiome or supplementation of specific bacterial genera/species in PAH. Fecal transplantation from angiotensin-converting enzyme 2 (Ace2)-overexpressing mice, which have less hypoxia-induced PH, to wild-type hypoxic mice attenuates PH development [26,43]. Administration of *Lactobacillus reuteri* to postnatal growth-restricted pups exposed to hyperoxia mitigates PH severity and RV hypertrophy [44]. *Lactobacillus rhamnosus* supplementation two weeks after MCT injection does not affect pulmonary vascular remodeling but enhances RV function [45].

Other approaches to modulate the microbiome involve human umbilical cord blood-derived mesenchymal stem cells (MSCs), which are nonhematopoietic cells that can self-renew and secrete antibacterial peptides [46]. There is recent interest in their ability to regulate the gut microbiome and alleviate inflammatory bowel diseases [47,48,49]. Treatment with MSCs rebalances the gut microbiome (reducing the disease-associated and increasing the anti-inflammatory bacteria) and attenuates hypoxia- [17] and MCT-induced PH [50,51]. Most of the animal studies modifying the gut microbiota in PH thus far use prevention models, intervening prior to or immediately after providing a stimulus to generate PH. Promising indirect evidence supports a need to further explore the role of the gut microbiome in PH. In SuHx mice, treatment with Ang1-7 four weeks after hypoxia initiation mitigates PH, partially attenuates disease-associated changes in gut microbiota, and enhances the beneficial metabolites, butyric acid and tryptophan [52]. Administering irbesartan, an angiotensin II receptor blocker, 30 days after starting exposure to hypoxia, mitigates PH, partially normalizes the *Firmicutes*-to-*Bacteroidetes* ratio, increases the intestinal abundance of *Lactobacillaceae* and *Lachnospiraceae*, and decreases *Prevotellaceae* and *Desulfovibrionaceae* in high-altitude PH hypobaric hypoxia rats [22]. Future studies need to evaluate the efficacy of altering the gut microbiome after PAH is established. Due to the many potential environmental confounders in clinical microbiome research, animal studies are needed to establish a robust foundation for biotherapeutics targeting the host microbiome composition and its systemic effects on respiratory health. Lastly, the potential gut–brain–lung axis in PAH [21,26,43] should be further explored.

**Table 1 idr-17-00066-t001:** Summary of preclinical studies linking the gut microbiome to pulmonary hypertension.

Study	Animal Model (PH Type)	Key Findings
Callejo et al. [15]	SuHx rats (PAH)	SuHx rats had increased *Firmicutes*-to-*Bacteroidetes* ratio (less abundant *Bacteroidetes* in PAH) Reduced acetate in serum of PAH rats
Hong et al. [16]	MCT rats (PAH)	Lower microbial diversity in PAH compared to healthy control Higher *Firmicutes*, *Proteobacteria*, and *Actinobacteria* and lower *Bacteroidota* and *Spirochaetota* in PAH Altered fecal metabolome in PAH Treatment of MCT rats with the calcium-sensing receptor antagonist NPS2143 (previously demonstrated to mitigate PAH severity) increased microbial diversity and reversed the fecal metabolite abnormalities
Luo et al. [17]	Hypoxia-induced mice (Group 3 PH)	Elevated *Firmicutes*-to-*Bacteroidetes* ratio in hypoxia-induced PH mice Decreased alpha diversity and richness in hypoxia-induced PH mice Mesenchymal stem cell injection mitigated hypoxia-induced PH and reversed the gut microbiota alterations
Cao et al. [18]	Hypoxia-induced rats (Group 3 PH)	Distinct gut microbiome and fecal metabolome between hypoxia-induced PH rats and normoxic controls Less intestinal microbial diversity and richness in hypoxia-induced PH
Luo et al. [19]	Hypoxia-induced rats (Group 3 PH), SuHx and MCT rats (PAH)	Dynamic alpha diversity values and richness indices during the development of hypoxia, SuHx, and MCT-induced PH
Chen et al. [20]	Left pulmonary artery ligation-induced PH rats (high flow-induced PH)	Altered gut microbiome (no difference in alpha diversity indices, decreased butyrate- and propionate-producing bacteria), gut metabolome (increased arginine), and lung metabolome between left pulmonary artery ligation-induced PH rat and sham-treated rats
Sharma et al. [21]	MCT rats (PAH)	Increased intestinal permeability, greater jejunum muscularis layer, and decreased villus length and number of goblet cells in MCT rats Altered gut microbiome in PAH (increased *Firmicutes*-to-*Bacteroidetes* ratio)
Nijiati et al. [22]	High-altitude hypobaric hypoxic rats (Group 3 PH)	Increased *Firmicutes*-to-*Bacteroidetes* ratio, decreased *Lactobacillaceae* and *Lachnospiraceae* abundance, and greater *Prevotellaceae* and *Desulfovibrionaceae* in ileocecal microbiome of high-altitude hypobaric hypoxic rats with reversal of these changes with irbesartan treatment
Adak et al. [23]	Hypobaric hypoxic rats (Group 3 PH)	Altered large intestinal microbial populations with hypoxia Necrotized large intestinal epithelial layer with greater lymphocyte infiltration in the lamina propria and reduced mucin-secreting goblet cells
Ranchoux et al. [24]	MCT rats (PAH)	More intestinal permeability (as assessed by measuring dextran-FITC, soluble CD14, and lipopolysaccharide) in MCT rats
Prisco et al. [25]	MCT rats (PAH)	Intermittent fasting MCT rats altered the gut microbiome, decreased right ventricular levels of microbiome metabolites (bile acids amino acid metabolites, and gamma-glutamylated amino acids), and augmented right ventricular function without altering PAH severity
Sharma et al. [26] Oliveira et al. [43]	ACE2 knock-in and wild-type (WT) hypoxic mice (Group 3 PH)	Global overexpression of ACE2 mitigated hypoxia-induced PH, neuroinflammation, pathological jejunal epithelium changes, and disruptions to the gut microbiome Fecal matter transfer from ACE2 knock-in mice attenuated hypoxia-induced PH
Gaowa et al. [27]	Heifers with brisket disease or high-altitude pulmonary hypertension (Group 3 PH)	Lower volatile fatty acids and alpha diversity in the rumen fluid of the bovine brisket disease model
Sanada et al. [29]	SuHx rats (PAH)	Antibiotic treatment (cocktail of ampicillin, vancomycin, neomycin, and metronidazole) of SuHx rats altered the gut microbiome and diminished pulmonary vascular remodeling
Pakhomov et al. [32]	Hypoxic mice (Group 3 PH)	Hypoxic mice fed a high-soluble-fiber diet had increased abundance of SCFA-producing bacteria, diminished proinflammatory bacteria, reduced interstitial macrophages, dendritic cells, and nonclassical monocytes in the lung, and attenuated PH severity
Karoor et al. [35]	Hypoxic rats (Group 3 PH)	Butyrate mitigated hypoxia-induced pulmonary hypertension, decreased alveolar and interstitial macrophages’ accumulation in the lungs, and upregulated tight junctional proteins in lung microvascular endothelial cells
Huang et al. [38]	MCT rats (PAH) and hypoxia-induced PH mice (Group 3 PH)	PH rats had higher circulating TMAO and treatment with DMB to inhibit TMAO synthesis reduced PH severity Administration of TMAO to hypoxia-induced PH mice worsened PH severity DMB suppressed macrophage production and proinflammatory cytokines/chemokines
Yang et al. [39]	MCT rats (PAH)	MCT rats treated with DMB had attenuated pulmonary vascular remodeling
Videja et al. [40]	MCT rats (PAH)	TMAO administration to MCT rats preserved right ventricular fatty acid oxidation, decreased pyruvate metabolism, and partially restored right ventricular function
Prisco et al. [45]	MCT rats (PAH)	*Lactobacillus* supplementation altered fecal micro/mycobiome, suppressed systemic inflammation, and enhanced right ventricular systolic and diastolic function without changing PAH severity
Abudukeremu et al. [52]	SuHx mice (PAH)	Treatment with Ang1-7 after hypoxia initiation mitigated PAH, augmented expression of intestinal occludin and ZO-1, partially attenuated disease-associated changes in gut microbiota, and elevated the beneficial metabolites, butyric acid and tryptophan
Wedgwood et al. [44]	Postnatal growth restriction rats (assigned to a larger liter at birth), exposed to hyperoxia	Postnatal growth restriction with or without hyperoxia altered distal small bowel and cecum microbiomes Treatment with the probiotic, *Lactobacillus reuteri* DSM, mitigated pulmonary hypertension and right ventricular hypertrophy and reduced alpha diversity
Marinho et al. [53]	*Schistosomiasis*-associated PAH mouse model	Greater gut microbiome alpha diversity in *Schistosomiasis*-associated PAH Increased *Firmicutes*-to-*Bacteroidetes* ratio and no change in relative abundance of *Deferribacteres* and *Proteobacteria*

ACE2: angiotensin-converting enzyme 2; Ang1-7: angiotensin (1-7); DMB: 3,3-dimethyl-1-butanol (inhibits TMAO synthesis); FITC: fluorescein isothiocyanate; MCT: monocrotaline; PAH: pulmonary arterial hypertension; PH: pulmonary hypertension; SCFA: short-chain fatty acids; SuHx: Sugen hypoxia; TMAO: trimethylamine-N-oxide; ZO-1: zonula occludens-1.

## 3. Clinical Evidence of the Gut–Lung Axis in PAH

Clinical studies show that PAH patients have gut dysbiosis and a leaky gut with increased bacterial translocation from the intestinal lumen to systemic circulation [24] (Table 2). Kim et al. [54] completed one of the initial clinical studies describing the distinct gut microbiome composition in PAH by studying the fecal microbiome of 18 PAH patients and 12 age- and sex-matched healthy controls. Compared to controls, PAH patients have a distinct microbiome composition with lower alpha diversity, fewer bacteria associated with polysaccharide fermentation and SCFA production (*Butyrivibrio crossotus*, *Bacteroides cellulosilyticus*, *Eubacterium siraeum*, *Bacteroides vulgatus*, *Akkermansia muciniphila*), and more bacteria associated with the proinflammatory metabolites, TMA and TMAO [54]. PAH patients also have a disparate intestinal virome [54].

In a subsequent single-center pilot study of 20 PAH patients and 20 healthy controls cohabiting with PAH patients (20 matched pairs), Jose et al. [55] observed no difference in alpha- (within a specific sample) or beta- diversity (between samples). In the largest PAH microbiota study to date of 72 patients, Moutsoglou et al. [56] demonstrated that the gut microbiome is less diverse in PAH patients compared to healthy controls and family members residing in the same household. Gut microbiome diversity correlates with measures of pulmonary vascular disease (mean pulmonary artery pressure, pulmonary vascular resistance, and pulmonary arterial compliance), but not with RV function [56], suggesting that the alterations in the gut microbiome are not due to RV failure and intestinal congestion. PAH patients have reduced abundance of gut bacteria containing genes encoding for the production of anti-inflammatory metabolites, specifically SCFAs (*Eubacterium ramulus*, *Firmicutes* sp. coabundance gene 110, *Coprococcus comes*, *Dorea longicatena*, *Bifidobacterium adolescentis*, *Gemmiger formicilis*, *Fusicatenibacter saccharivorans*, *Eubacterium hallii*, *Anaerostipes hadrus*, *Gordonibacter pamelaeae*, *Ruminococcus torques*, *Coprococcus catus*, *Coprococcus eutactus*, and *Blautia obeum*) and secondary bile acids (*Collinsella aerofaciens*, *Coprococcus eutactus*, *Anaerostipes hadrus*, *Eubacterium ramulus*, *Blautia obeum*, *Eubacterium hallii*, *Ruminococcus bicirculans*, *Ruminococcus torques*, *Eubacterium eligens*, *Fusicatenibacter saccharivorans*, *Roseburia faecis*, *Dorea longicatena*, *Coprococcus catus*, and *Roseburia hominis*) and increased relative abundance of bacteria with genes encoding for the production of TMAO (*Clostridium bolteae*, *Escherichia coli*, and *Klebsiella pneumoniae*) [56]. Consistent with the above gut microbiome changes, PAH patients have relatively lower circulating levels of anti-inflammatory microbial metabolites (SCFA and bile acids) and a trend towards higher levels of the proinflammatory microbial metabolite TMAO. In a study of 35 patients with IPAH [38] and another of 124 PAH patients [39], TMAO levels are elevated in higher-risk patients, suggesting that TMAO is associated with worse outcomes. The role of the other intestinal microorganisms (fungi, protozoans, and archaea) in clinical PAH is not well-established.

Environmental effects, such as high altitude and hypoxia, may also alter the gut microbiome. A small study of six highlander PH patients living on the Tibetan plateau range and seven lowlander PH patients (residents of Shanghai) demonstrated that while there are overall divergent gut microbial signatures between PH patients and controls, altitude contributes to the gut microbiota differences [57]. TMA synthesis enzymes are enriched in lowlanders with PH, and there is no difference in TMA-producing microbiota between highlander controls and highlander PH patients [57].

Whether there is a causal relationship between the gut microbiome/metabolites and PAH is not yet determined in clinical studies. Mendelian randomization of data from the MiBioGen consortium, the largest genome-wide meta-analysis of intestinal microbiota [58], investigated the potential direct link between gut microbiota, metabolites, diet, and PAH [59]. The bacteria, *Alistipes* and *Victivallis*, correlate with increased PAH risk, while *Coprobacter*, *Erysipelotrichaeae*, *Lachnospiraceae*, and *Ruminococcaceae* protect against PAH [59]. However, SCFAs, TMAO, and dietary patterns are not causally associated with PAH in a Mendelian randomization analysis [59]. In another recent Mendelian randomization study, Su et al. identified 11 gut microbial taxa, including *Bifidobacteriaceae*, *Eubacterium eligens* group, and *Sutterella*, and 24 bacterial metabolites that are linked to PAH pathogenesis by regulating the expression of *ITPR2*, *IDE*, *NRIP1*, and *IGF1* genes in lung tissue [60]. Limitations of these studies are that the design only evaluates the impact of genetic factors on intestinal microbiota abundance and the development of PAH, a small effect size with the use of single-nucleotide polymorphisms (SNPs) for metabolites and bacteria, Mendelian randomization being prone to false positives, and cohorts primarily of European ethnicity, restricting the generalizability of the findings. Thus, more clinical studies are needed to determine whether there is a casual connection between the intestinal microbiota/metabolites and PAH.

There are now emerging data showing that not only is the gut microbiome changed in PAH but also the airway microbiome composition may be distinct. In a study of PH patients of various etiologies (Group 1 PAH, PH due to lung disease, and chronic thromboembolic PH), Zhang et al. [61] observed higher alpha-diversity and increased *Streptococcus, Lautropia*, and *Ralstonia* in the airways of PH patients compared to reference controls. Intratracheal instillation of *Streptococcus* induced PH in rats [62]. A recent study identified disparate airway mycobiomes or fungal compositions between PH and healthy controls [63]. Further research should assess the microbiome/mycobiome in other parts of the body.

**Table 2 idr-17-00066-t002:** Summary of the clinical studies delineating the role of the gut microbiome in PAH.

Study	Cohort	Key Findings
Ranchoux et al. [24]	21 healthy controls, 19 idiopathic PAH patients, 22 heritable PAH patients carrying a *BMPR2* mutation	Idiopathic and heritable PAH patients had increased serum soluble CD14 levels compared to healthy controls Untreated severe PAH patients (cardiac index < 3 L/min/m^2^) had higher serum lipopolysaccharide levels compared to treated severe PAH patients.
Kim et al. [54]	18 PAH patients, 12 age- and sex-matched healthy reference subjects	PAH patents had altered gut microbiome with increased bacterial communities associated with TMA/TMAO and purine metabolism and decreased butyrate- and propionate-producing bacteria
Jose et al. [55]	20 PAH patients, 20 healthy controls (PAH subject simultaneously enrolled with cohabitating non-PAH control subject)	No difference in microbial abundance or diversity (alpha diversity, beta diversity, or *Firmicutes*-to-*Bacteroidetes* ratio)
Moutsoglou et al. [56]	72 PAH patients, 15 family control subjects residing within the same household as a PAH patient, 39 healthy controls	PAH patients had less diverse gut microbiome, lower plasma SCFAs and secondary bile acids, and enrichment of microbial genes that encoded TMA Shannon diversity index correlated with PAH severity but not right ventricular function
Huang et al. [38]	35 idiopathic PAH patients, 19 age- and sex-matched healthy controls	Circulating TMAO was elevated in intermediate- to high-risk PAH patients compared to low-risk PAH patients and healthy controls
Yang et al. [39]	124 PAH patients (40 idiopathic/heritable PAH, 82 PAH associated with congenital heart disease, and 2 with PVOD)	High plasma TMAO was associated with worse WHO functional class, elevated N-terminal pro-brain natriuretic peptide, and reduced cardiac index
Dong et al. [57]	13 PH patients (46% highlanders), 88 controls (70% highlanders)	Distinct gut microbial composition in PH patients compared to controls No difference in alpha diversity (Shannon index) between PH patients and controls TMA species or synthesis enzymes were more enriched in lowlanders with PH. Among highlanders, this discrepancy was not seen between PH patients and controls

*BMPR2*: bone morphogenetic protein receptor type 2; PAH: pulmonary arterial hypertension; PH: pulmonary hypertension; PVOD: pulmonary veno-occlusive disease; SCFAs: short-chain fatty acids; TMA: trimethylamine; TMAO: trimethylamine N-oxide; WHO: World Health Organization.

## 4. Potential Approaches to Modulate the Gut–Lung Axis to Treat PAH

There are several different strategies to restructure the microbiome, including diet/prebiotics, probiotics, postbiotics [64] (inanimate microorganisms or their components), microbiota transplant, medications, vaccines [65], exercise [66], and mesenchymal stromal cell therapy [51,67] (Figure 3). There have not yet been many clinical trials investigating how altering the gut microbiome affects PAH. There is an ongoing study evaluating the impact of microbiota transplant from healthy controls to PAH patients [68]. Unfortunately, the SARS-CoV-2 pandemic impeded the enrollment and initiation of the trial assessing the use of chlorhexidine mouthwash and oral nitrate therapy in PH patients (NCT03787082). Challenges to translating preclinical findings to PAH patients include genetic factors, age, sex, concomitant chronic diseases, and different environmental exposures [69] (medications, diet, timing of eating, toxin use, chemicals in the environment/products, hygiene, air pollution, daylight exposure, etc.). Fetal/maternal or perinatal microbiota exposures may also complicate the efficacy of microbiota clinical trials. For example, supplementing omega-3 polyunsaturated fatty acids in the diet of pregnant rats improves RV systolic pressure and survival in pups exposed to hyperoxia at the time of birth [70]. Additionally, the timing and duration of gut microbiome modulation needed to confer benefits in PAH are unknown. Despite these possible confounders, the vast potential health benefits of altering the microbiome in PAH should be explored to a greater extent.

## 5. Contribution of Infections in PAH Pathogenesis

Infectious agents, including bacteria, viruses, fungi, and parasites, can cause pulmonary arterial injury and inflammation by impacting vascular cells, leading to severe pulmonary vascular remodeling and PAH [71]. The pathophysiology and mechanisms by which infections lead to pulmonary vascular disease are more extensively reviewed elsewhere [72]. Infections in the setting of a dysregulated immune system may elevate susceptibility to PH development [73,74]. There is now growing interest in determining how host microbiota promote or resist infections [75].

A common global cause of PAH is infection by the intravascular parasite, *Schistosoma mansoni*. *Schistosomiasis* affects over 200 million people with about 1–10 million chronically infected people at risk of developing PAH [76,77]. *Schistosomiasis* cases occur in many regions worldwide, but the majority are observed in Africa and Asia, although longstanding epidemiological and socioeconomic challenges may underestimate its global impact. *Schistosomiasis* disrupts both gut and lung microbiota [53]. *S. mansoni* egg exposure decreases lung alpha diversity, mainly by impacting the relative abundance of the phylum, *Ascomycota*, while the pulmonary *Firmicutes*-to-*Bacteroidetes* ratio remains unchanged [53]. In contrast, *Schistosomiasis* increases gut microbiota alpha diversity and the *Firmicutes*-to-*Bacteroidetes* ratio [53], suggesting significant differences in lung and gut microbiome responses to *Schistosomiasis* infection.

HIV infection is also a well-recognized cause of PAH [78]. Host factors and geography contribute to the fecal microbiota disruptions that occur after HIV infection [79]. Geographic location has a greater effect on fecal microbiota composition than HIV infection status [79]. Interestingly, while HIV infection is known to disrupt gut epithelial barrier function, there are regional differences in immune activation with elevated soluble CD14 levels in HIV-infected individuals from all three regions studied (United States, Botswana, and Uganda), but there is increased intestinal fatty acid-binding protein in HIV-infected individuals from only the United States and Botswana [79]. Thus, distinct gut microbial alterations due to the host region are major confounders in microbiota studies.

Unquestionably, additional work is needed to unravel the precise mechanisms contributing to the complex interactions between infections and host microbiota. Future studies should further ascertain whether microbiome alterations after infection impact susceptibility to PAH.

## 6. Future Directions

There is a need to explore whether modifying the gut microbiome is effective in combating established PAH and to define the approaches (e.g., diet, gut metabolites, and/or specific microorganisms) that are advantageous in PAH. Moreover, future research should assess the potential role of the other gut microorganisms (fungi, viruses, archaea, protozoa, bacteriophages, etc.) in PAH pathogenesis. Additional knowledge gaps include how environmental exposures affect the gut microbiome, whether the microbiomes of other tissues (e.g., lung, skin, etc.) impact PAH pathogenesis, and the role of the microbiome in other PH subtypes.

## 7. Conclusions

Animal and clinical studies reveal an altered intestinal microbiota composition in PAH. Modulating the gut microbiome diminishes systemic inflammation and pulmonary immune cell infiltration. Additional research is needed to delineate whether restructuring the microbiome attenuates PAH after it is established and to explore the role of the gut and airway/lung microbiome in infection-associated PAH.

## Figures and Tables

**Figure 1 idr-17-00066-f001:**
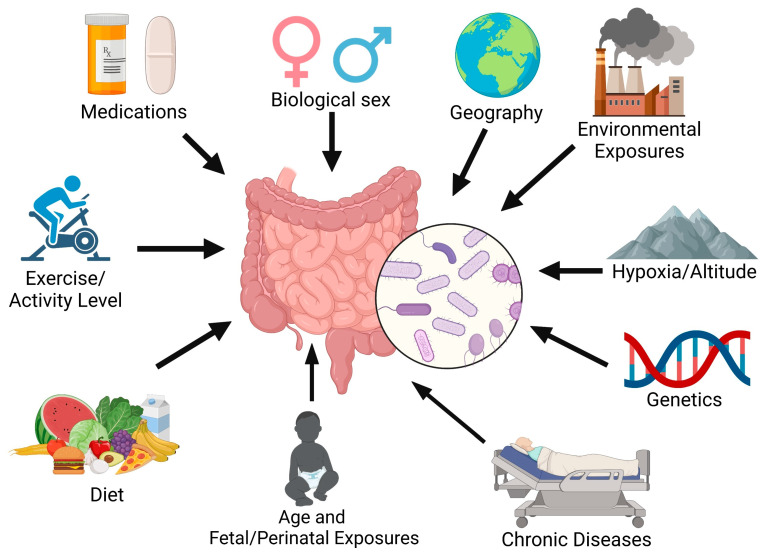
Factors influencing intestinal microbiome composition.

**Figure 2 idr-17-00066-f002:**
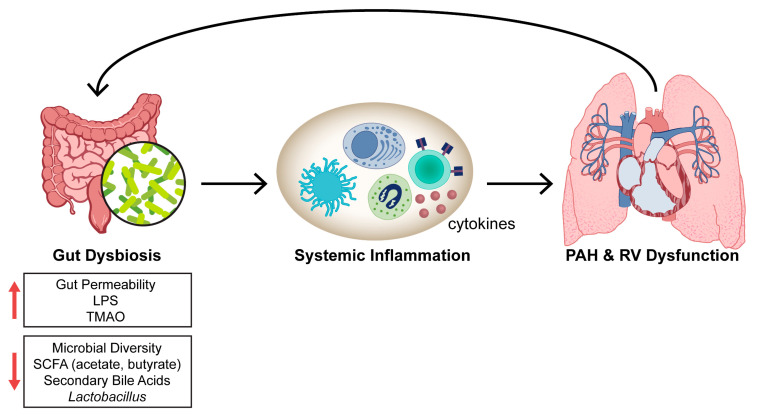
Working hypothesis linking gut dysbiosis to PAH and/or RV dysfunction. LPS: lipopolysaccharide; PAH: pulmonary arterial hypertension; RV: right ventricle; SCFA: short-chain fatty acids; TMAO: trimethylamine N-oxide.

**Figure 3 idr-17-00066-f003:**
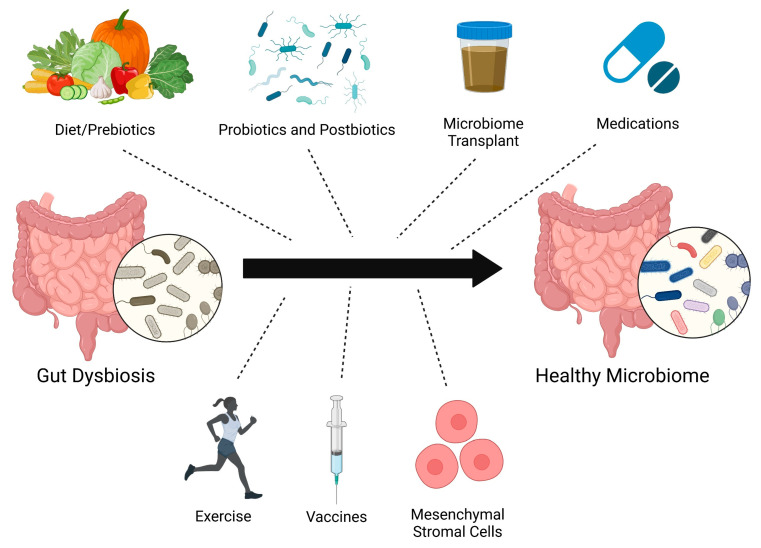
Potential therapeutic approaches to restructure the gut microbiome.

## Data Availability

No new data were created or analyzed in this study.

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
