# Peer review of "Gut Microbiome in Pulmonary Arterial Hypertension—An Emerging Frontier"

_2036-7449, 2025, doi:10.3390/idr17030066_

Round 1

Reviewer 1 Report

Comments and Suggestions for Authors

I congratulate the authors for their work. It is an interesting subject. The manuscript is well organised, containing information from both animal and human studies. It is comprehensive.

Here some comments:

-the abstract is too short, does not provide enough background information  to stimulate curiosity to read the article; please develop it -  add information about pulmonary hypertension and microbiome;

-please add the methods – I did not find in the manuscript information about how the articles were selected, which databases were used; please add databases used for research, keywords used...

- there is a lot of information, hard to follow, I would suggest two tables summarising sections 1 and 2 would make it easier to follow and understand the text;

- before the conclusions section, based on the research work done, please add future directions of research;

Author Response

I congratulate the authors for their work. It is an interesting subject. The manuscript is well organised, containing information from both animal and human studies. It is comprehensive.

Here some comments:

  1. The abstract is too short, does not provide enough background information to stimulate curiosity to read the article; please develop it - add information about pulmonary hypertension and microbiome;

We thank the reviewer for this feedback. We have expanded the abstract to include essential background information on pulmonary hypertension and the emerging role of the microbiome.

  1. Please add the methods – I did not find in the manuscript information about how the articles were selected, which databases were used; please add databases used for research, keywords used...

We appreciate the reviewer’s comments. Typically, in a review manuscript, methods are not included. We found and selected articles by searching PubMed, Google, and Google Scholar by searching “pulmonary arterial hypertension”, “pulmonary hypertension”, and/or “microbiome”. As this approach for searching for manuscripts to include in a review article is standard and review articles do not have a designated methods section, if possible, we would like to omit adding this information.

  1. There is a lot of information, hard to follow, I would suggest two tables summarising sections 1 and 2 would make it easier to follow and understand the text;

Thank you for this feedback. We have added two tables (Tables 1 and 2) to our manuscript, one summarizing the preclinical data and the other summarizing the clinical data linking the gut microbiome to PAH.

  1. Before the conclusions section, based on the research work done, please add future directions of research.

We thank the reviewer for this suggestion. We have added a Future Directions section to our review article.

Reviewer 2 Report

Comments and Suggestions for Authors

This is relatively new are and the authors presented a lot of investigations and summarize the current data in this field. However, since this is review article for wider interest, I suggest to authors to write 3 tables, for the first 3 sections, where they will present the short design and results of the most important studies in the field. 

Please explain precisely what is alpha and beta diversity since you mentioned this several times through the manuscript and what are Ace, Sobs and Simpson indices which measure that diversity. 

Two figures describe the potential role of factors to microbiota and the second potential therapeutic approach on microbiome. However, the most important schema is not presented, the potential links between microbiota and pulmonary hypertension. 

One important comment, there are several types of PH, and the majority of investigations presented are on type 1 PH, eventually type IV. It is important to comment this in the introduction and conclusion. 

Author Response

  1. This is relatively new area and the authors presented a lot of investigations and summarize the current data in this field. However, since this is review article for wider interest, I suggest to authors to write 3 tables, for the first 3 sections, where they will present the short design and results of the most important studies in the field. 

Thank you for this feedback. We have added two tables to the text to summarize the preclinical and clinical data supporting the role of the microbiome in PAH.

  1. Please explain precisely what is alpha and beta diversity since you mentioned this several times through the manuscript and what are Ace, Sobs and Simpson indices which measure that diversity. 

In the second paragraph of the “Clinical Evidence of the Gut-Lung Axis in PAH” section, we describe alpha diversity as diversity within a specific sample while beta diversity is diversity between samples.

Common indices to assess alpha diversity include Sobs, ACE, and Simpson. Sobs (observed species) simply counts the number of species or operational taxonomic units (OTUs) detected in a sample, providing a direct measure of richness. ACE (Abundance-based Coverage Estimator) also estimates species richness but gives greater weight to rare species, making it a more sensitive estimator for under-sampled communities. Simpson’s Index, on the other hand, emphasizes evenness by measuring the probability that two individuals randomly selected from a sample will belong to the same species, with lower values indicating greater diversity and more even species distribution. Shannon’s Index is also common and accounts for both richness and evenness in a single value, placing greater emphasis on species richness and therefore, capturing the uncertainty in predicting the species identity of a randomly chosen individual.

As the technical details of determining ACE, Sobs, and Simpson/Shannon index are beyond the scope this review and may lead to confusion, we have removed reference to these different indices that assess alpha diversity. Removal of this text does not impact the key points of the review article as we do not reference these specific indices anywhere else in the text.

  1. Two figures describe the potential role of factors to microbiota and the second potential therapeutic approach on microbiome. However, the most important schema is not presented, the potential links between microbiota and pulmonary hypertension. 

We thank the reviewer for this recommendation. We have created the below figure delineating the potential link between the gut microbiota and pulmonary arterial hypertension and right ventricular dysfunction.

  1. One important comment, there are several types of PH, and the majority of investigations presented are on type 1 PH, eventually type IV. It is important to comment this in the introduction and conclusion. 

Thank you for this feedback. There are five World Health Organization (WHO) groups of pulmonary hypertension (PH) and PAH is Group 1 PH. We mostly focus on WHO group 1 PH or PAH but also describe some studies supporting the role of the microbiome in other types of PH (e.g. PH due to hypoxia which is known as Group 3 PH). We have added the following to the end of the introduction: “While there are five broad World Health Organization (WHO) groups of pulmonary hypertension (PH), which are defined by a mean pulmonary artery pressure ≥20 mmHg[14], we primarily focus on WHO group 1 PAH and Group 3 PH (PH secondary to lung disease) in this review.”

We now also describe in the newly added Future Directions section the need to further define the role of the microbiome in other PH types.

Reviewer 3 Report

Comments and Suggestions for Authors

The review article presents a timely and comprehensive review of the emerging role of gut microbiome in pulmonary arterial hypertension (PAH), which is gaining appreciation in the recent literature. The authors have integrated an extensive amount of literature from animal models, clinical studies, and microbiome-related metabolites. The topic is highly relevant and of great scientific interest with evident translational value.

Here are my concerns regarding the review.

Major Comments:

  1. While the review is very thorough, there is lack of well-framed hypothesis. Writing the content around a framework will enhance clarity and structure. May be a figure depicting the interaction of gut-lung and their impact on PAH progression will be helpful too.
  2. Although microbial metabolites like SCFAs and TMAO are well covered but fundamental molecular mechanism linking these metabolites to inflammation, vascular remodeling, and heart dysfunction are not sufficiently included.
  3. Inclusion of clinical trial going on related to the study along with practical challenges to translating microbiome-targeted therapies can also be discussed.

Minor Comments:

  1. Is PH and PAH different? If not, please be consistent throughout the article.

Author Response

The review article presents a timely and comprehensive review of the emerging role of gut microbiome in pulmonary arterial hypertension (PAH), which is gaining appreciation in the recent literature. The authors have integrated an extensive amount of literature from animal models, clinical studies, and microbiome-related metabolites. The topic is highly relevant and of great scientific interest with evident translational value.

Here are my concerns regarding the review.

Major Comments:

  1. While the review is very thorough, there is lack of well-framed hypothesis. Writing the content around a framework will enhance clarity and structure. May be a figure depicting the interaction of gut-lung and their impact on PAH progression will be helpful too.

Thank you for this suggestion. At the end of the introduction, we have added the following text: “The overarching hypothesis is that gut dysbiosis leads to increased systemic inflammation and progression of PAH and/or RV dysfunction (Figure 2).” Furthermore, we have now added the below figure (Figure 2) to depict the possible interaction of the gut microbiome and PAH pathogenesis.

  1. Although microbial metabolites like SCFAs and TMAO are well covered but fundamental molecular mechanism linking these metabolites to inflammation, vascular remodeling, and heart dysfunction are not sufficiently included.

We thank the reviewer for this comment. In our review, we summarize the data linking the microbial metabolites, SCFAs and TMAO, to inflammation in pulmonary arterial hypertension. Buytrate’s anti-inflammatory (PMIDs: 28654658, 31759015, 32476236, 28802151, 24226770) and TMAO’s pro-inflammatory effects (PMIDs: 26903003, 28871042, 28153917) have also have demonstrated in other disease states. At this time, the microbiome research field has not clearly delineated all of the fundamental mechanism(s) linking these metabolites to inflammation (and subsequently to vascular remodeling and cardiac dysfunction) and we believe that this discussion is beyond the scope of our review.

  1. Inclusion of clinical trial going on related to the study along with practical challenges to translating microbiome-targeted therapies can also be discussed.

In the “Potential Approaches to Modulate the Gut-Lung Axis to Treat PAH”, we discuss the ongoing clinical microbiota transplant trial and describe the challenges of translating microbiome-targeted therapies: “Challenges to translating preclinical findings to PAH patients include genetic factors, age, sex, concomitant chronic diseases, and different environmental exposures68 (medications, diet, timing of eating, toxin use, chemicals in environment/products, hygiene, air pollution, daylight exposure, etc.). Fetal/maternal or perinatal microbiota exposures may also complicate the efficacy of microbiota clinical trials”

Minor Comments:

  1. Is PH and PAH different? If not, please be consistent throughout the article.

Thank you for this feedback. There are five World Health Organization (WHO) groups of pulmonary hypertension (PH) and PAH is Group 1 PH. We mostly focus on WHO group 1 PH or PAH but also describe some studies supporting the role of the microbiome in other types of PH (e.g. PH due to hypoxia which is known as Group 3 PH). We have added the following to the end of the introduction: “While there are five broad World Health Organization (WHO) groups of pulmonary hypertension (PH), which are defined by a mean pulmonary artery pressure ≥20 mmHg[14], we primarily focus on WHO group 1 PAH and Group 3 PH (PH secondary to lung disease) in this review.”

Additionally, we have carefully gone through our manuscript to mention “PAH” when discussing Group 1 PH and “PH” when considering other types of PH.

Round 2

Reviewer 3 Report

Comments and Suggestions for Authors

Thank you for your thorough revisions. The authors have addressed all comments thoughtfully and the manuscript has improved significantly.